# An Update on Toll-like Receptor 2, Its Function and Dimerization in Pro- and Anti-Inflammatory Processes

**DOI:** 10.3390/ijms241512464

**Published:** 2023-08-05

**Authors:** Katrin Colleselli, Anna Stierschneider, Christoph Wiesner

**Affiliations:** Department of Medical and Pharmaceutical Biotechnology, IMC University of Applied Sciences, 3500 Krems, Austria

**Keywords:** Toll-like receptor 2, dimerization, homodimerization, heterodimerization, inflammation, neuroinflammation, neurodegeneration

## Abstract

While a certain level of inflammation is critical for humans to survive infection and injury, a prolonged inflammatory response can have fatal consequences. Pattern recognition Toll-like receptors (TLRs) are key players in the initiation of an inflammatory process. TLR2 is one of the most studied pattern recognition receptors (PRRs) and is known to form heterodimers with either TLR1, TLR4, TLR6, and TLR10, allowing it to recognize a wide range of pathogens. Although a large number of studies have been conducted over the past decades, there are still many unanswered questions regarding TLR2 mechanisms in health and disease. In this review, we provide an up-to-date overview of TLR2, including its homo- and heterodimers. Furthermore, we will discuss the pro- and anti-inflammatory properties of TLR2 and recent findings in prominent TLR2-associated infectious and neurodegenerative diseases.

## 1. Introduction

Over the past few decades, a great deal of knowledge has been gained about human innate and adaptive immunity and its key players, including the pattern recognition receptors (PRRs). These germline-encoded receptors can rapidly identify the specific molecular structures on the surface of pathogens, so-called pathogen-associated molecular patterns (PAMPs) or endogenous damage-associated molecular patterns (DAMPs), linking non-specific immunity to specific immunity [1,2]. To date, two distinct classes of PRRs have been described: the membrane-bound receptors, including Toll-like receptors (TLRs) and C-type lectin receptors (CLRs), and the cytoplasmic proteins, such as nucleotide-binding oligomerization domain (NOD)-like receptors (NLRs), retinoic acid-inducible gene (RIG)-I-like receptors, absent in melanoma 2 (AIM2)-like receptors (ALRs), and DNA or RNA sensors, such as cyclic GMP-AMP synthase (cGAS) [3,4,5]. TLRs are type I transmembrane proteins with an extracellular domain containing leucine-rich repeats that facilitate recognition of PAMPs. They also have a transmembrane domain and an intracellular toll-interleukin-1 receptor (TIR) domain needed for downstream signaling [6]. So far, ten receptors (TLR1-10) have been identified in the human body, which can be divided into cell surface receptors such as TLR1, TLR2, TLR4-6, TLR10, and intracellular receptors including TLR3, and TLR7-9 [7,8]. TLRs are widely distributed on cells of the immune system, such as monocytes, as well as on non-immune cells, such as endothelial or epithelial cells [9]. Upon PAMP or DAMP recognition, homo- or heterodimerization of TLRs is triggered, leading to an intracellular signaling cascade that in most cases results in a pro-inflammatory response [10]. A prominent example of homodimerization is TLR4, which is induced after exposure to lipopolysaccharide (LPS) isolated from the outer membrane of Gram-negative bacteria such as *Escherichia coli*, whereas TLR2 is known to form heterodimers with either TLR1 or TLR6, depending on the ligand [11,12]. In addition to heterodimer formation, TLR2 can function as a single receptor and in a homodimer, albeit the latter is still under debate [12]. Although TLRs, especially TLR2, have been extensively studied, their role in health and disease is still very much in the spotlight, as evidenced by the large number of studies on TLRs and SARS-CoV-2 infection, for example [13,14,15].

In this review, we aim to provide an up-to-date overview of TLR2 dimerization, including homo- and heterodimerization, and propose a model to further investigate these dimerization mechanisms. Additionally, the expanded role of TLR2 with its pro- and anti-inflammatory capabilities, and the recent findings in prominent TLR2-associated (neuro) inflammatory diseases will be discussed.

## 2. TLR2 Ligands and Signaling

TLRs, which are the mammalian orthologues of the *Drosophila melanogaster* Toll receptor originally discovered in 1985, are well conserved and primarily associated with pattern recognition [12,16,17]. TLR2, along with TLR4, is one of the most studied pattern recognition receptors, and unlike certain other receptors in this family, TLR2 is known to form various heterodimers with TLR1, TLR4, TLR6, and TLR10 [18,19,20]. In recent years, a growing number of studies have shown that TLR2 may be the most versatile TLR, as this receptor recognizes a broad repertoire of ligands from a variety of pathogen sources and interacts with a large number of other receptors [12,21].

TLR2/1 heterodimers can sense triacylated lipopeptides (LPs) from Gram-negative bacteria or mycoplasma, such as lipoarabinomannans and lipomannans, whereas TLR2/TLR6 heterodimers recognize diacylated LPs, including lipoteichoic acid (LTA) from Gram-positive bacteria and mycoplasma [12,22,23,24]. Depending on the ligand, low-endotoxic atypical LPS can induce TLR2/TLR4 heterodimerization, and the TLR2/TLR10 heterodimer has been found to participate in *Helicobacter pylori* LPS recognition [25,26]. Additional co-receptors, such as clusters of differentiation 14 (CD14), have been described that bring CD14 and TLR2 and TLR1 into physical proximity by binding LPs to induce signaling [27]. Interestingly, the mRNA expression of TLR2, but not TLR1, TLR4, and TLR6, was strongly induced in alveolar macrophages by prolonged LPS treatment (24 h) [28]. Besides heterodimerization, some evidence also points to the existence of TLR2 homodimerization, but this is still controversially discussed [12,29]. Commonly used TLR2 ligands for in vitro and in vivo studies are synthetic di- and triacylated LPs such as Pam2CSK4, Pam3CSK4, isolated from *Escherichia coli*, and FSL-1, which represents a pathogen-associated molecular pattern of the LP secluded from *Mycoplasma fermentans* or *Mycoplasma salivarium* [12,30,31]. Moreover, endogenous ligands, danger signals for TLR2, have been identified, including heat shock proteins, human β-defensin-3, high mobility group box 1 protein (HMGB1), and hyaluronan fragments [32,33,34,35]. Other ligands of TLR2 that play a major role in neurodegenerative disorders are amyloid-β (Aβ) and α-synuclein (αSyn) [36,37]. Therefore, in addition to PAMPs, endogenous TLR2 activation by host-derived danger signals is a viable factor in the pathogenesis of inflammation and related diseases.

Ligand-induced dimerization is the starting point for TLR2 signal transduction. Dimerization brings the TIR domains of the cytoplasmic tails into close proximity, providing a platform for signaling by TIR domain-containing adaptor molecules [12]. TLR2 primarily relies on the myeloid differentiation factor 88 (MyD88) and the Toll-interleukin-1 receptor domain-containing adaptor protein (TIRAP) adaptors for signaling [38]. The MyD88-dependent pathway involves death-domain interactions mediating intracellular signaling in a stepwise manner. Specifically, MyD88 recruitment is indirect and mediated by TIRAP [39]. Active MyD88 then sequentially triggers the phosphorylation of interleukin-1 receptor-associated kinase 4 (IRAK4), IRAK1, and IRAK2 [40]. The IRAK complex engages tumor necrosis factor (TNF) receptor-associated factor 6 (TRAF6), which undergoes K63-linked autoubiquitination and ubiquitinates nuclear factor kappa B (NF-κB) essential modulator (NEMO). The complex of transforming growth factor-β-activated kinase-1 (TAK1), TAK1-binding protein 2 (TAB2), and TAB3 is then activated. Next, TAK1 phosphorylates IkappaB kinase alpha (IKKα) and IKKβ, and the IKKs phosphorylate IκB, are marked for degradation, which ultimately leads to the production of pro-inflammatory cytokines via NF-κB and activating protein-1 (AP-1), and activation of mitogen-activated protein kinase (MAPK), modulating cell proliferation and survival (Figure 1) [6,41]. Interestingly, the interaction of TIRAP with IRAK1 and IRAK4 can lead to the degradation of TIRAP following its phosphorylation and ubiquitination, resulting in the inhibition of TLR2 signaling [42].

## 3. Dimerization and Additional Co-Receptors of TLR2

### 3.1. Homodimerization

The existence of TLR2 homodimers has been mentioned in a number of publications in recent years, however, the function of this homodimer is still controversial [12,43,44,45]. In vitro studies such as those of de Groot et al. suggest that TLR2 is functional as a homodimer, although it may require co-receptors to reach an active state. In particular, they were able to show that TLR2 signaling was reduced in the TLR1- and TLR6-deficient reporter systems, but was still evident in response to *Mycoplasma pneumoniae* and *Streptococcus pneumoniae* [46]. In a different investigation, another research group concluded that LTA-activated TLR2 via the TLR2 homodimer is likely to be less potent than that of LPs via the TLR2 heterodimer, which is limited to the induction of IRAK-M [47]. Su et al. identified diprovocim-1 as an effective inducer not only for the formation of TLR2/TLR1 heterodimers, but also for the formation of TLR2 homodimers in vitro [48]. Contrary studies show that the proline-proline-glutamic acid 18 (PPE18) protein of *Mycobacterium tuberculosis* induces TLR2 homodimerization, which triggers anti-inflammatory type responses that cause increased activation of the mammalian p38 MAPK [49]. Furthermore, by generating a CRISPR/Cas9-mediated knock-out of endogenous TLR6 in JE6-1 TLR2/6 reporter cells, it was recently shown that TLR2 homodimers are expressed on the cell surface. However, these reporter cells did not respond to any of the bacterial TLR2 agonists tested, nor could they be detected by diprovocim-1 induction. Activation of reporter cells occurred only in cells expressing TLR2/1 heterodimers [50]. Taken together, the recent results suggest that the function of a TLR2 homodimer is strongly dependent on ligands and co-receptors yet to be discovered. To address these unanswered questions, novel methods and strategies could help to further elucidate potential functions of the TLR2 homodimer.

### 3.2. TLR2 Heterodimers

The ability of TLR2 to dimerize not only with itself but also with a variety of TLRs, greatly expands the spectrum of detectable pathogens. Ozinsky et al. were among the first to discover that TLR2 relies on heterodimerization with either TLR1 or TLR6 to initiate adequate cell activation and a pro-inflammatory response [18,21]. Structural studies of the heterodimers have supported the importance of ligand binding to stabilize TLR2/1 and TLR2/6 dimerization for downstream signaling [51,52]. However, since the heterodimerization of TLR2 with TLR1 or TLR6 has already been extensively studied, we will focus on insights into other TLR2 heterodimerizations. There is accumulating evidence that TLR10 is capable of forming functional heterodimers with TLR2 [20,53]. This is plausible because in mammals, TLR10 is a member of the TLR1 subfamily and shares numerous paralogous genes with TLR1, 2, and 6 [54,55]. To assess whether the TLR2/10 heterodimer induces downstream signaling, Pachathundikandi et al. examined the mRNA expression levels of pro-inflammatory cytokines upon exposure to *Helicobacter pylori* or LPS and found a significant upregulation of interleukin-1β (IL-1β) in TLR2/10-transfected HEK293 cells [56]. In another study, they silenced TLR10 in the human colon adenocarcinoma cell line HT-29 and in monocytic THP-1 cells, which led to increased viability of *Listeria monocytogenes*, and they unveiled that the heterodimer of TLR2/TLR10 was the one behind the activation of NF-κB [57]. Nagashima et al. further confirmed this by using NC1-N87 gastric cells incubated with *Helicobacter pylori* and concluded that the TLR2/TLR10 heterodimer upregulates NF-κB activation to a greater extent than other TLR2 heterodimers [26]. A recent research report indicated that TLR10 regulates TLR2-induced cytokine production in monocytes isolated from Parkinson’s disease patients [58]. However, the specific ligand(s) and function of the TLR2/10 heterodimer have not been fully elucidated. Interestingly, a report from 2014 suggested that the hemoglobin-induced TLR2/4 heterodimer mediates inflammatory injury in intracerebral hemorrhage in in vivo and in vitro models [19]. Furthermore, an in vitro study by Muniz-Bongers et al. showed that in HEK293, matrix metalloproteinase 2 (MMP2) binds to both TLR2 and TLR4 for signaling, suggesting that TLR2 and TLR4 may form a heterodimer and consequently trigger a MyD88-dependent signaling cascade [59]. The existence of functional TLR2/4 heterodimers was further confirmed by Francisco et al. who showed that low-endotoxic atypical LPS isolated from *Ochrobactrum intermedium* is a potent TLR2/TLR4-inducing agonist, which they additionally confirmed by molecular docking analysis and fluorescence resonance energy transfer [25]. Nevertheless, the detailed mechanism of these divergent TLR2 heterodimers is still in question, and additional research and specific ligands are needed to substantiate these findings.

### 3.3. Other Co-Receptors Supporting TLR2

Additional co-receptors, called accessory receptors, are often required to augment TLR2 ligand delivery, pattern recognition, and other functions (see Figure 2) [12,21,60]. Prominent adaptors include CD14 and CD36, which are glycoproteins that are expressed primarily on monocytes and macrophages and that promote TLR2-dependent inflammation [61,62,63,64,65]. By performing immunoprecipitation and sugar inhibition assays, the physical interaction of ArtinM, a TLR2 agonist, with TLR2 and CD14 was observed to be a prerequisite for ArtinM-induced M1 macrophage activation [66]. Another study demonstrated that treatment of the LPS-incubated human microglia with anti-CD36 regulated the inflammatory cytokine levels in the brains of newborn mice and resulted in a significant decrease in TLR2 expression levels, while TLR4 expression was not altered [67]. In addition to CD14 and CD36, the hyaluronan receptor CD44 was found to be significantly upregulated in THP-1 wild-type versus TLR2 knock-out cells using quantitative mass spectrometry [68]. A number of reports in the last few years have indicated an interaction between TLR2 and CD44 [69,70,71,72]. Another interesting group of TLR2 co-receptors are the heterodimeric integrin receptors, which are critical for facilitating cell–cell and cell–extracellular matrix adhesion during inflammation [73]. They are composed of an α and a β subunit, which allows them to recognize a variety of different ligands that are relevant to cell adhesion and migration [74]. Numerous research articles report that β1, β2, as well as β3, directly or indirectly interact with TLR2 to initiate signaling cascades that culminate in cytokine secretion [21,68]. An in vitro study from 2019 demonstrated that α- 1- acid glycoprotein potentiated TLR2-dependent β2-integrin-mediated cell adhesion of neutrophils [75]. On the other hand, De Baros et al. showed that Pam3CSK4 increased α3β1 integrin levels, which was inhibited by silencing TLR2. However, *Paracoccidioides brasiliensis* infection led to an increase in TLR2 and in parallel, α3β1 integrin levels were downregulated in human lung epithelial cells [76]. These results once again emphasize that TLR2 modulation is ligand- and cell-type dependent.

### 3.4. Optogenetics, an Innovative Approach to Study TLR Dimerization and Signaling

Conventional techniques to further identify and characterize dimerization and signaling mechanisms, such as pharmacological control including ligand exposure, knock-out models, or biochemical assays, may not always be the most appropriate method. For example, models based on TLR2 knock-outs emphasize the importance of TLR2 but do not provide insight into the detailed signaling mechanism or interactions with other co-receptors. However, new experimental models, such as optogenetics, are emerging that do not rely on ligand binding to activate cellular receptors. Optogenetics was first introduced by Deisseroth in 2006 in the field of neuroscience, where light-sensitive ion channels were being utilized to remotely control action potentials and thereby neuronal networks [77]. In short, this biological technology uses genetic engineering to combine an effector protein with a light-sensitive protein domain of microbial, fungal, or plant photoreceptors to precisely control the activity of specific cells with light [78]. To enable the activation and inactivation of signaling pathways, a variety of optically controlled receptors have been developed in recent years, including G protein-coupled receptors (GPCRs) or receptor tyrosine kinases (RTKs) [79,80,81,82]. Grusch et al. have used the light–oxygen–voltage (LOV) sensing domain to induce RTK dimerization by light [80]. In a variety of eukaryotic and prokaryotic proteins, LOV domains bind a flavin chromophore to function as blue light sensors. In order to use these LOV domains for optogenetic approaches, they were isolated from different organisms, including *Arabidopsis thaliana* or *Chlamydomonas reinhardtii* [83,84]. Our research group has established light-inducible TLR4 cell models using human pancreatic adenocarcinoma and endothelial cell lines [85,86]. For this purpose, the LOV domain, isolated from the yellow-green alga *Vaucheria frigida aureochrome 1*, was C-terminally fused to the full-length TLR4. This innovative system allows precise on/off switching under temporal and spatial control, which may be an interesting approach to elucidate TLR2 homodimerization and signaling.

## 4. TLR2 Mediates Both Pro- and Anti-Inflammatory Responses

TLR2 activation can evoke pro-inflammatory and anti-inflammatory responses that are highly dependent on the cell type and the cell compartment in which it is expressed, the ligand, and the co-receptors. A complete overview of all cells expressing TLR2 and the functional impact of TLR2 activation is beyond the scope of this review. However, to illustrate the diverse and complex effects of TLR2 activation, a selection of recent findings will be mentioned in this section. While the well-known heterodimerization of TLR2 with TLR1 or TLR6 leads to the classical pro-inflammatory response in different cell types, the anti-inflammatory reaction serving as an immune regulator seems to be more heterogeneous. A study from 2021 showed that a high number and blockwise distribution of certain methyl esters of pectins were responsible for a TLR2/1-dependent anti-inflammatory effect [87]. Interestingly, the flavonoid baicalin isolated from *Scutellaria baicalensis Georgi* inhibited inflammation in rats with chronic obstructive pulmonary disease via the TLR2/MyD88/NF-κB pathway [88]. Furthermore, the anti-inflammatory properties of TLR2 were evident in response to polysaccharide A (PSA) from *Bacteroides fragilis* in B cells and T cells, resulting in the production of IL-10 and interferon-γ (IFN-γ) [89,90,91]. M2 macrophages from healthy donors or patients with rheumatoid arthritis showed reduced anti-inflammatory activity in the presence of abundant TLR2 [92]. Soluble TLR2 (sTLR2), which is generated by proteolytic cleavage of the TLR2 transmembrane protein, also known as ectodomain shedding, has been suggested as a promising biomarker of infections and systemic inflammation [93]. In this context, it has been shown that soluble and full-length TLR2 are released by macrophages under anti-inflammatory conditions, implicating that vesicle-bound full-length TLR2 exerts decoy properties that may be involved in immune suppression [28].

Although pattern recognition appears to be the most common activity of TLR2, it is not its only biological function. Our recent publication indicated that TLR2 also plays an important role in cell adhesion and migration of monocytes [68]. More specifically, we performed several functional cell-based assays using wild-type, TLR2 knock-out, and TLR2 knock-in THP-1 cells. We found that TLR2 enhances quicker and stronger adhesion of monocytes to the endothelium and a more pronounced endothelial barrier disruption after endothelial activation. The importance of ligand-activated TLR2 in cell adhesion and migration has been shown previously, but we pointed out that activation of endothelial cells is sufficient for a distinct effect of TLR2 expressed on monocytes [94,95,96]. In this context, several studies have confirmed the involvement of TLR2 in coagulation, resulting in increased monocyte adhesion, upregulation of tissue factor, increased extrinsic pathway coagulation including active recruitment of platelets, and formation of platelet–monocyte aggregates [97,98,99]. Interestingly, TLR2 was also found to be expressed in early endosomes, late endosomes/lysosomes, and Rab-11-positive compartments but not in the Golgi apparatus or endoplasmic reticulum of monocytes [100]. Recent data have provided evidence that IL-10 secretion in response to *Listeria monocytogenes* infection in vitro and in vivo (mouse model) is generally TLR2 dependent and that immune suppression by phagosome-confined bacteria in vivo is mostly dependent on endosomal TLRs [101]. Table 1 lists the most recent findings of pro- and anti-inflammatory responses induced by TLR2.

## 5. The Role of TLR2 in Infectious and Neurodegenerative Diseases

TLR2 and its signaling have been implicated in a variety of inflammatory diseases such as rheumatoid arthritis, inflammatory bowel disease, asthma, and multiple sclerosis [106,107,108,109]. However, here we will discuss sepsis and COVID-19, both of which can lead to chronic inflammation and, in addition, neuroinflammation that may promote neurodegenerative diseases such as Alzheimer’s disease (AD) or Parkinson’s disease (PD) (see Figure 3).

### 5.1. Sepsis

Sepsis is described as a systemic dysregulated immune response of the body to an infectious reaction caused by pathogens, including bacteria, viruses, fungi, and parasites [110,111]. This life-threatening syndrome damages the body’s tissues and organs, ultimately leading to organ failure and death, with mortality rates reaching 20–70% if left untreated [112,113]. Advances in medical health care have made it possible for most patients to survive the initial hyperinflammatory phase of sepsis [114]. However, the transition to the later immunosuppressive phase is still critical, with 30% of patients dying from secondary infections [115,116]. In recent years, TLR2 and TLR4, which are found on immune cells such as monocytes, endothelial cells, or platelets, have gained attention in the context of human sepsis [117,118]. Given the inherent role of TLR2 in acute and chronic inflammation, TLR2 needs to be strictly controlled to avoid a dysregulated host response. If this regulation fails, sepsis can be triggered by pathogens that activate platelets, which then interact with endothelial cells and immune cells, leading to disruption of the endothelial barrier, fluid leakage, and ultimately to tissue damage and organ failure [116]. Another scenario describes the activation of the endothelium by pathogens, resulting in the release of chemokines and cytokines (cytokine storm) that trigger the recruitment of platelets and immune cells, ultimately leading to a hyperinflammatory response that causes apoptosis, endothelial barrier breakdown, fluid leakage, and further tissue damage [119]. It has been shown that *Streptococcus pneumoniae* induces platelet activation via TLR2 and that its inhibition completely abolished platelet aggregation, implicating TLR2 in the thrombotic complications of sepsis [120]. Another research group studied 59 patients with *Staphylococcus aureus* bacteremia and found that TLR2 downregulation and high IL-6 and IL-10 levels, indicative of immune dysregulation during early bacteremia, may be linked to mortality [121]. Furthermore, Hoppstädter et al. showed that TLR2 is highly upregulated during the immunosuppressive phase of systemic inflammatory response syndrome (SIRS) and sepsis patients [28]. Consistent with previous data, using the experimental model of polymicrobial sepsis induced by cecal ligation and puncture, one group showed increased TLR2 levels in the kidney and intestine [122]. In the same sepsis mouse model, another group has recently demonstrated that TLR2-deficient mice have lower levels of IL-10 and reduced caspase-3 activation in the spleen [123]. Taken together, these findings underscore the multifaceted role of TLR2 in the hyperinflammatory state and also in sepsis-induced immunosuppression.

### 5.2. COVID-19

Severe acute respiratory syndrome coronavirus 2 (SARS-CoV-2), which has challenged human health and public safety in recent years, is a member of the *Coronaviridae* family of β-coronaviruses, which are positive-stranded, enveloped RNA viruses [124]. Clinical features include an increase in inflammatory monocytes and neutrophils, and a robust inflammatory milieu that includes cytokines such as IL-1β, IL-6, and TNF-α [125,126]. While these cytokines are an inherent part of innate immunity and essential for the elimination of pathogens, uncontrolled release can trigger hyperinflammation, leading to cell death and tissue damage [127,128]. Zheng et al. demonstrated that TLR2 can recognize the SARS-CoV-2 envelope (E) protein and is required for inflammatory cytokine release during β-coronavirus infection. In addition, they found that SARS-CoV-2 E protein can induce TLR2-dependent inflammation in the lungs of mice [129]. Consistent with this, another study also showed that E protein–TLR2 interaction leads to the downstream activation of the transcription factor NF-κB, which stimulates the production of IL-8 [130]. However, it has also been proposed that prophylactic intranasal administration of a TLR2/6 agonist reduces SARS-CoV-2 transmission and provides protection against COVID-19 [131]. Notably, sepsis and COVID-19 patients share many pathophysiological and clinical features, such as thrombocytopenia, coagulopathy, vascular micro thrombosis, high cytokine production, septic shock, multiorgan dysfunction syndrome, fever, leukopenia, hypotension, or leukocytosis [132]. These data highlight the importance of temporal dependency in the context of TLR2, which can lead to protective or harmful effects.

### 5.3. Neuroinflammation

TLR2 is expressed not only on a variety of immune cells, but also on neuronal and glial cells in the central, peripheral, and enteric nervous systems, enabling them to act as immune cells [133,134]. As evidenced by numerous publications in recent years, TLR2 appears to play a pivotal role in neurological and neurodegenerative disorders by influencing brain cells and plasticity. Neuroinflammation is characterized by functionally activated and proliferating microglia and astrocytes and increased levels of pro-inflammatory cytokines, chemokines, and cytotoxic molecules [135,136,137]. Ultimately, this leads to permeabilization of the blood–brain barrier, infiltration of peripheral immune cells, and neuronal cell death [137,138,139]. Although neuroinflammation is an essential tool to protect the central nervous system, a persistent inflammatory response, also known as chronic inflammation, can lead to the damage and death of neurons, predisposing to neurodegenerative diseases such as AD or PD [140,141].

#### 5.3.1. Alzheimer’s Disease

The most prevalent neurodegenerative disease in the aging population, AD, is pathologically characterized by the accumulation of Aβ plaques, hyperphosphorylated tau in neurofibrillary tangles, and consequently, enhanced neuroinflammation [142,143]. Since inflammation has a significant impact on the pathogenesis of AD, several inflammatory mediators have been proposed as AD markers, including IL-1β, TNF-α, NF-κB, and TLR2 [144]. A large body of research shows that inflammation can cause Aβ or tau overload, which then triggers inflammatory responses, creating a downward spiral of neuroinflammation and pathology [144,145,146]. In particular, microglia, the main cellular component of the innate immune system in the brain, express high levels of TLR2, making them a valuable target in the initiation, progression, and preservation of Alzheimer’s disease [147,148]. Together with CD14 and TLR4, TLR2 is essential for fibrillar Aβ-stimulated microglial activation [149]. Rangasamy et al. showed that selectively targeting the TLR2–MyD88 interaction rescued AD pathology, including hippocampal glial activation and reduced Aβ burden [150]. However, in the context of TLR2, there is a fine line between acute, protective inflammation and chronic, neurotoxic inflammation, and the challenge is to find the right framework for therapeutic approaches.

#### 5.3.2. Parkinson’s Disease

PD is the most prominent form of α-synucleinopathies and the second most common neurodegenerative disease [151]. The pathological hallmark of PD is Lewy bodies, which are protein-containing inclusions in which misfolded and aggregated αSyn is the major component [152,153]. Similar to AD, the spread and process of αSyn misfolding remains elusive but in vivo approaches along with a variety of in vitro studies have provided strong evidence that environmental and genetic factors, as well as inflammation, are key contributors [154,155]. Since TLR2 can recognize so-called alarmins, including αSyn, which are released from damaged neuronal cells and lead to the production of cytokines, other immune cells, such as T cells and B cells, can be recruited to stimulate an adaptive immune response [156]. It has been suggested that TLR2 is upregulated in the pathogenesis of synucleinopathies in the brains of PD patients, thereby establishing TLR2 as a mediator of not only pro-inflammatory but also neurotoxic effects of extracellular α-synuclein aggregates. Kim et al. showed that blocking TLR2 reduced αSyn accumulation in neuronal and astroglial cells, neuroinflammation, neurodegeneration, and functional deficits in a PD mouse model [157]. Consistent with the previous study, it has recently been demonstrated that neuronal TLR2 activation acutely impairs the lysosomal autophagy pathway and amplifies αSyn pathology seeded with α-Syn preformed fibrils using human cell models. In addition, TLR2 knock-out in a human neuroblastoma cell line and the use of a small molecule TLR2 inhibitor (NPT1220-321) in induced pluripotent stem cell-derived neurons from a patient with PD have both demonstrated that αSyn pathology can be attenuated [158]. Another promising study has shown that specific inhibition of the TLR2 interaction domain of MyD88 and the NEMO binding domain can αSyn propagation in vitro and in vivo [159]. Taken together, several studies implicate TLR2 in the pathogenesis of PD but it is still unclear how exactly TLR2 affects the αSyn pathology [160].

## 6. Conclusions and Future Perspectives

One of the major discoveries in medicine over the last twenty years has been that the immune system and subsequent inflammatory processes are associated not only with a few specific diseases but also with a broad range of mental and physical health conditions [161,162]. In fact, chronic inflammatory disorders are now considered the leading cause of death worldwide, with more than 50% of all deaths due to inflammation-related diseases such as stroke, cancer, ischemic heart disease, autoimmune diseases, neurodegenerative diseases, etc. [163,164]. A key player in these inflammation-mediated events is TLR2, one of the most extensively studied pattern recognition receptors [12,18]. A considerable number of in vitro and in vivo studies have been conducted to fully elucidate the role of this protein. Among all known TLRs, TLR2 is rather exceptional in the sense that heterodimerization with multiple TLRs is possible, broadening the ligand spectrum and also exhibiting pro- and anti-inflammatory capacities [12,28]. Evidence has accumulated that TLR2 signaling is triggered by SARS-CoV-2, inducing robust pro-inflammatory cytokine expression that may contribute to severe COVID-19 [165]. Moreover, TLR2 has been identified as a key effector of neuroinflammation, which is highly relevant to the pathophysiology of neurodegenerative diseases [166]. Despite these advances, there are still many gaps in our knowledge of TLR2 mechanisms in health but particularly in disease. We have only scratched the surface of the mechanistic link between TLRs and neurodegenerative diseases, for example. Clearly, more studies, especially with patient-derived samples, are needed to further assess the complexity. In addition, emphasis should be placed on in silico techniques which provide attractive tools not only for drug design and discovery methodologies but also for cellular models of various pathologies [167,168]. There is an urgent need to fill these gaps in order to develop therapeutic strategies for inflammation-induced diseases, including neurodegenerative diseases, which will become an even greater burden as the population ages. As highlighted in a number of recent reports, neurological disorders pose a major challenge to the sustainability of global health care [169,170,171]. In this review, we have attempted to provide a current view of TLR2 focusing on dimerization, optogenetics as a novel experimental model, and involvement in common diseases which may help to push new and innovative ideas on its way.

## Figures and Tables

**Figure 1 ijms-24-12464-f001:**
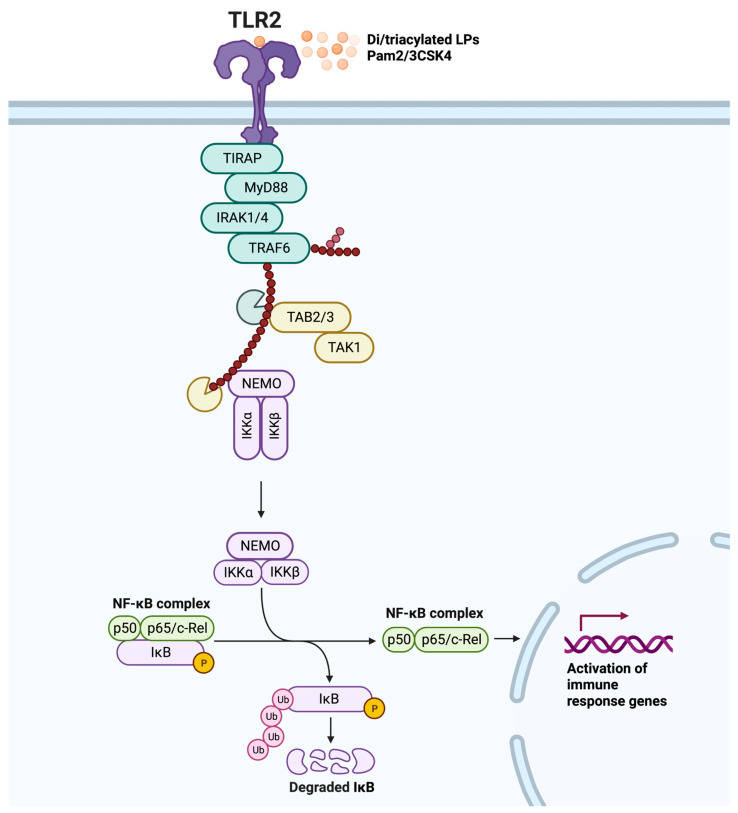
TLR2 MyD88-dependent signaling pathway. Dimerization is triggered by ligand binding (diacylated lipopeptides (LPs) or triacylated LPs, and synthetic Pam2CSK4 or Pam3CSK4), resulting in a signaling cascade that begins with Toll-interleukin-1 receptor domain-containing adaptor protein (TIRAP) binding to TLR2, which then leads to the recruitment of MyD88. Following phosphorylation of interleukin-1 receptor-associated kinase (IRAK) 4, IRAK1, and IRAK2, TNF receptor-associated factor 6 (TRAF6) undergoes K63-linked autoubiquitination and ubiquitinates nuclear factor kappa B (NF-κB) essential modulator (NEMO). The complex of transforming growth factor-β-activated kinase-1 (TAK1), TAK1-binding protein 2 (TAB2), and TAB3 complex is then activated to phosphorylate IkappaB kinase alpha (IKKα) and IKKβ, and the IKKs phosphorylate IκB, marking it for degradation and releasing the NF-κB complex (consisting of p50 and p65/c-Rel). This ultimately leads to the activation of immune response genes, including the production of pro-inflammatory cytokines via the transcription factor NF-κB. Adapted from “NF-KB signaling pathway”, by BioRender.com (2023).

**Figure 2 ijms-24-12464-f002:**
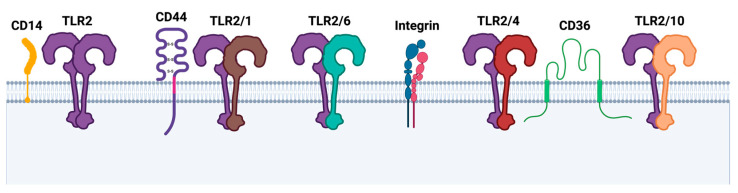
TLR2 homo- and heterodimers and additional co-receptors. Upon ligand binding, dimerization is initiated, often requiring additional co-receptors such as cluster of differentiation 14 (CD14), CD44 receptor, integrin receptors, or CD36. Created with BioRender.com.

**Figure 3 ijms-24-12464-f003:**
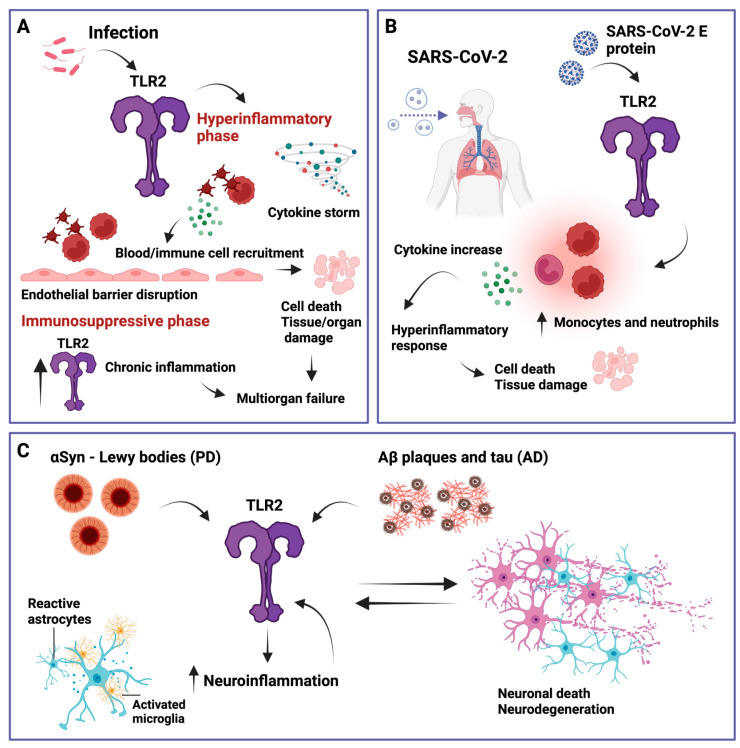
Simplified overview of TLR2 involvement in several relevant inflammatory diseases. (**A**) Sepsis. When immune regulation fails and a hyperinflammatory state is induced via elevated cytokine production (cytokine storm), sepsis can be triggered by TLR2, which further activates platelets and other immune cells that then interact with endothelial cells, leading to endothelial barrier disruption, cell death, and tissue and organ damage. In addition, TLR2 has been shown to be elevated during the immunosuppressive phase of sepsis which can lead to multiorgan failure. (**B**) Severe acute respiratory syndrome coronavirus 2 (SARS-CoV-2). TLR2 can recognize the SARS-CoV-2 envelope (E) protein, leading to an increase in inflammatory monocytes and neutrophils, inducing a hyperinflammatory response that can cause cell death and tissue damage. (**C**) Parkinson’s disease (PD) and Alzheimer’s disease (AD). TLR2 can recognize α-synuclein (αSyn) and amyloid-β (Aβ) plaques and tau, respectively. This promotes neuroinflammation characterized by reactive astrocytes and activated microglia, which further increase TLR2 levels, leading to a feedback loop of neuronal cell death, further TLR2 upregulation, and neuroinflammation. Created with BioRender.com.

**Table 1 ijms-24-12464-t001:** TLR2-dependent pro- and anti-inflammatory responses.

TLR Dimer	Ligand	Cytokines/Chemokines	References
TLR2/1	Triacyl LPs Pam3CSK4	IL-6, IL-8, TNF-α, IL-1α, IL-1β, IL-12, IL-13, IFN-α, IFN- β	[12,102,103]
TLR2/6	Diacyl LPs Pam2CSK4 FSL-1	IL-6, IL-8, IL-12, TNF-α, IFN-α, IFN- β, CXCL-10	[12,104,105]
TLR2/?	*Listeria monocytogenes*	IL-10	[101]
TLR2/?	*Bacteroides fragilis* PSA	IL-10, IFN-γ	[89,90,91]
TLR2/?	LPS	TNF, CXCL10, IL10	[28]
TLR2/4	Hemoglobin, atypical LPS	IL-12, IL-6, TNF-α	[25]
TLR2/10	*Helicobacter pylori*	TNF- α, IL-1β	[56]

## Data Availability

Not applicable.

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
