# Peer review of "An Update on Toll-like Receptor 2, Its Function and Dimerization in Pro- and Anti-Inflammatory Processes"

_ijms, 2023, doi:10.3390/ijms241512464_

Round 1
Reviewer 1 Report
Dear Authors,
This paper addresses an interesting topic, in which you are acquainted; and there are no any fundamental errors. The review is well written, well organized, the introduction is not as extended as it could be, but the remaining sections are good enough. Only some editorial changes are required (table 1 needs wider spaces between lines, as well as references font should be edited according to the template).
Best regards and good luck
Dear Editors,
First of all, I would like to thank you for the possibility of reviewing this article. This paper is in the field of International Journal of Molecular Sciences. It addresses an interesting topic. I recommend publishing this review.
-
Areas of strength:
-
Well organized and comprehensive
-
English used correct and readable
-
Interest to the readers may be high, which high potential citations
-
Well present material and methods as well as results
-
Authors have acquaintance with this field
Your sincerely
Author Response
Comment: This paper addresses an interesting topic, in which you are acquainted; and there are no any fundamental errors. The review is well written, well organized, the introduction is not as extended as it could be, but the remaining sections are good enough. Only some editorial changes are required (table 1 needs wider spaces between lines, as well as references font should be edited according to the template).
Response: Thank you very much for your feedback and suggestions! We have added more details to the introduction: A prominent example of homodimerization is TLR4, which is induced after exposure to lipopolysaccharide (LPS) isolated from the outer membrane of Gram-negative bacteria such as Escherichia coli, whereas TLR2 is known to form heterodimers with either TLR1 or TLR6, depending on the ligand [11,12]. In addition to heterodimer formation, TLR2 can function as a single receptor and in a homodimer, whereby the latter is still under debate [12]. Although TLRs, especially TLR2, have been extensively studied, their role in health and disease is still very much in the spotlight, as evidenced by the large number of studies on TLRs and SARS-CoV-2 infection, for example [13–15].
We also added more spacing to Table 1 and re-examined the references.
Reviewer 2 Report
According to Katrin Colleselli and colleagues' manuscript entitled "An Update on Toll-Like Receptor 2, its Function and Dimerization in Pro- and Anti-Inflammatory Processes". Although a certain level of inflammation is necessary for humans to survive infection and injury, a prolonged inflammatory response can be fatal. Pattern recognition Toll-like receptors (TLRs) play an important role in the initiation of an inflammatory response. It is known that TLR2 forms heterodimers with either TLR1, TLR4, TLR6, and TLR10, allowing it to recognize a wide range of pathogens. There have been many studies conducted over the past decades concerning TLR2 mechanisms in health and disease, but there are still many questions that remain unanswered. This review provides an up-to-date overview of TLR2, including its homodimers and heterodimers. The pro-inflammatory and anti-inflammatory properties of TLR2 will also be discussed, as well as recent findings in prominent TLR2-associated infectious diseases and neurodegenerative conditions. Regarding the present manuscript, I would like to make a few comments.
-The introduction should provide more detailed information about TLR2's role in different aspects.
I believe that the information in the main text in Figure 2 is incomplete. Perhaps more information about ligands and examples of their use in anti- or pro-inflammatory conditions would be helpful in the main text.
-What is the advantage of including neurodegenerative diseases rather than other immunological conditions?
Is COVID19 on the same line?
As I read the review, it seems to provide a good explanation; however, I have a simple question. Does TLR2 have any relationship with the microbiota?
Author Response
Comment 1: The introduction should provide more detailed information about TLR2's role in different aspects.
Response to comment 1: Thank you for this suggestion. We added more information about TLR2 to the introduction: A prominent example of homodimerization is TLR4, which is induced after exposure to lipopolysaccharide (LPS) isolated from the outer membrane of Gram-negative bacteria such as Escherichia coli, whereas TLR2 is known to form heterodimers with either TLR1 or TLR6, depending on the ligand [11,12]. In addition to heterodimer formation, TLR2 can function as a single receptor and in a homodimer, whereby the latter is still under debate [12]. Although TLRs, especially TLR2, have been extensively studied, their role in health and disease is still very much in the spotlight, as evidenced by the large number of studies on TLRs and SARS-CoV-2 infection, for example [13–15].
Comment 2: I believe that the information in the main text in Figure 2 is incomplete. Perhaps more information about ligands and examples of their use in anti- or pro-inflammatory conditions would be helpful in the main text.
Response to comment 2: We have added more details about ligands to Chapter 2. TLR2/1 heterodimers can sense triacylated lipopeptides (LPs) from Gram-negative bacteria or mycoplasma, such as lipoarabinomannans and lipomannans, whereas TLR2/TLR6 heterodimers recognize diacylated LPs, including lipoteichoic acid from Gram-positive bacteria and mycoplasma [12,22–24]. Moreover, endogenous ligands, danger signals, for TLR2 have been identified, including heat shock proteins, human β-defensin-3, high mobility group box 1 protein (HMGB1), and hyaluronan fragments [32–35]. Therefore, in addition to PAMPs, endogenous TLR2 activation by host-derived danger signals is a viable factor in the pathogenesis of inflammation and related diseases.
Furthemore, we have added some more recent studies of the pro-and anti-inflammatory capacities of TLR2 to the manuscript A study from 2021 showed that a high number and blockwise distribution of certain methyl esters of pectins were responsible for a TLR2/1-dependent anti-inflammatory effect [87]. Interestingly, the flavonoid baicalin isolated from Scutellaria baicalensis Georgi in-hibited inflammation in rats with chronic obstructive pulmonary disease via the TLR2/MyD88/NF-κB pathway [88].
Comment 3: What is the advantage of including neurodegenerative diseases rather than other immunological conditions?
Response to comment 3: This is true, there are many other immunological diseases. However, this would exceed the scope of this review. Since the first author has a background in neuroscience and the connection between neurodegenerative diseases and immunology has gained attention in recent years, we decided to focus on that as well.
Comment 4: Is COVID19 on the same line?
Response to comment 4: COVID19 has been very present in the last few years and has many similarities to sepsis and chronic inflammatory conditions. Since there have been a lot of studies on TLR2, we thought it was worth mentioning.
Comment 5: As I read the review, it seems to provide a good explanation; however, I have a simple question. Does TLR2 have any relationship with the microbiota?
Response to comment 5: Thank you very much for your feedback and this interesting question. Due to its potential as a regulator or modifier of human health, disease, and therapeutic response, the microbiome has become a highly interesting target for research. Furthermore, autoimmune diseases and neurodegenerative diseases have been linked to a dysregulated microbiome. There are some interesting studies regarding TLRs and TLR2 related to the microbiome: Wasko et al., 2020, doi: 10.1016/j.autrev.2019.102430; Yiu et al., 2017, doi: 10.1007/s00109-016-1474-4.
We have added an introducing passage to chapter 5. TLR2 and its signaling have been implicated in a variety of inflammatory diseases such as rheumatoid arthritis, inflammatory bowel disease, asthma and multiple sclerosis [106–109]. However, here we will discuss sepsis and COVID19, both of which can lead to chronic inflammation and, in addition, neuroinflammation that may promote neuro-degenerative diseases such as Alzheimer's disease (AD) or Parkinson's disease (PD).
Reviewer 3 Report
In this narrative review, various molecular aspects (e.g. dimerization effect) and physiological effects in immunological homeostasis promoted by Toll-Like Receptor 2 are exposed in an orderly and in-depth manner. references are ≥10 and old) is arranged in 5 sections described comprehensively. Some minor changes (form and content) will improve the scientific value of the manuscript.
· Syntax and grammar should be reviewed once again.
· Do not forget to describe the meaning of abbreviations the first time they are mentioned. If possible, reduce them as much as possible.
· Reduce the preamble statements and feed with relevant data from the material reviewed.
· All figures should be provided with a much higher resolution (>300 dpi). The figures should be self-explanatory, so it is suggested to further feed their corresponding footnotes.
· Although the proportion of new and old references is adequate, many of them are not properly formatted for this journal.
· Where needed, the contribution of this new manuscript to knowledge on the subject should be highlighted, comparing it with what was previously published (see doi: 10.1038/s41577-021-00577-0, 10.1111/sji.12771, 10.4110/in. 2021.21.e18, 10.1155/2021/1157023)
Moderate editing of English language required
Author Response
Comment 1: Syntax and grammar should be reviewed once again.
Response to comment 1: Thank you for this remark. We have carefully re-evaluated the entire manuscript.
Comment 2: Do not forget to describe the meaning of abbreviations the first time they are mentioned. If possible, reduce them as much as possible.
Response to comment 2: We have reviewed all abbreviations once again, thank you.
Comment 3: Reduce the preamble statements and feed with relevant data from the material reviewed.
Response to comment 3: We have taken this comment into account and changed some passages as for instance: However, these are not the only heterodimers that TLR2 is capable of forming. Depending on the ligand, low-endotoxic atypical LPS can induce TLR2/TLR4 heterodimerization and the TLR2/TLR10 heterodimer has been found to participate in Helicobacter pylori LPS recognition [25,26]. Much research has been conducted to shed light on the interaction of TLR2 with CD14 and CD36 [61–63]. CD14 and CD36 are glycoproteins expressed primarily on monocytes and macrophages that promote TLR2-dependent inflammation [64,65].
Comment 4: All figures should be provided with a much higher resolution (>300 dpi). The figures should be self-explanatory, so it is suggested to further feed their corresponding footnotes.
Response to comment 4: We have added the highest possible resolution to all figures and added additional information to the figure descriptions.
Comment 5: Although the proportion of new and old references is adequate, many of them are not properly formatted for this journal.
Response to comment 5: Thank you for your comment, we have reviewed and formatted all references again.
Comment 6: Where needed, the contribution of this new manuscript to knowledge on the subject should be highlighted, comparing it with what was previously published (see doi: 10.1038/s41577-021-00577-0, 10.1111/sji.12771, 10.4110/in. 2021.21.e18, 10.1155/2021/1157023)
Response to comment 6: Thanks for these valuable references. We have taken a closer look and added some of the references to the manuscript.
Round 2
Reviewer 2 Report
Thank you for taking into account my previous comments. Following my reading of the article, I do not feel it necessary to make any further comments.